# Hoxa5 Inhibits the Proliferation and Induces Adipogenic Differentiation of Subcutaneous Preadipocytes in Goats

**DOI:** 10.3390/ani12141859

**Published:** 2022-07-21

**Authors:** Dingshuang Chen, Yaqiu Lin, Nan Zhao, Yong Wang, Yanyan Li

**Affiliations:** 1College of Animal Science and Veterinary, Southwest Minzu University, Chengdu 610041, China; dingshuangchen@163.com (D.C.); linyq1999@163.com (Y.L.); xiaozhao202208@163.com (N.Z.); wangyong010101@swun.cn (Y.W.); 2Key Laboratory of Qinghai-Tibetan Plateau Animal Genetic Resource Reservation and Utilization, Ministry of Education, Southwest Minzu University, Chengdu 610041, China; 3Key Laboratory of Sichuan Province for Qinghai-Tibetan Plateau Animal Genetic Resource Reservation and Exploitation, Southwest Minzu University, Chengdu 610041, China

**Keywords:** Hoxa5, adipogenesis, proliferation, goat, preadipocyte

## Abstract

**Simple Summary:**

The homeobox a5 (Hoxa5) is not only a critical developmental transcription factor, but also plays an important role in the regulation of adipocyte differentiation and lipid metabolism. In our study, we found that Hoxa5 promotes goat subcutaneous preadipocyte adipogenic and inhibits its proliferation. This provides a theoretical basis for the regulation of goat subcutaneous preadipocyte differentiation and proliferation.

**Abstract:**

The homeobox a5 (Hoxa5) plays considerable roles in the differentiation and lipid metabolism of adipocytes. However, the current knowledge about the mechanistic roles and functions of Hoxa5 in goat subcutaneous preadipocyte remains unclear. Therefore, Hoxa5 loss-of-function and gain-of-function was performed to reveal its functions in adipogenesis. For differentiation, overexpression of Hoxa5 notably increased the expression of adipogenic genes (PPARγ, CEBP/α, CEBP/β, AP2, and SREBP1), as well as promoted goat subcutaneous preadipocyte lipid accumulation. Knockdown of Hoxa5 mediated by siRNA technique significantly inhibited its differentiation and suppressed the accumulation of lipid droplets. Regarding proliferation, overexpression of Hoxa5 reduced the number of cells stained with crystal violet, and inhibited mRNA expression of the marker genes including CCNE1, PCNA, CCND1, and CDK2, and also significantly reduced EdU-positive rates. Consistently, knockdown of Hoxa5 demonstrated the opposite tendency. In conclusion, these data demonstrated that Hoxa5 promotes adipogenic differentiation of goat subcutaneous preadipocyte and inhibits its proliferation in vitro.

## 1. Introduction

Adipose tissue is an important multifunctional organ, which not only affects the health of the human, but also affects the meat quality of meat-producing animals [1,2,3] In humans, many diseases are closely related to adipose tissue, comprising of hypertension, type 2 diabetes, obesity, cardiovascular disease, and so on [4,5,6,7,8,9,10]. In domestic animals, however, the distribution of adipose tissue affects the meat flavor and carcass quality [11,12,13]. Thus, unraveling the metabolic features and regulatory mechanism of adipose tissue development is of both enormous medical value and economic value. In this study, we isolated goat subcutaneous preadipocytes as an in vitro experimental model to study the potential molecular mechanisms of preadipocyte adipogenesis. Transcription factors (TFs) are prominent components of gene transcriptional regulation, as they inhibit or activate the expression of target genes by specifically recognizing and binding to the TFs binding site in the promoter region [14,15]. TFs have been demonstrated to serve a critical role during fat deposition. Emerging evidence has found that TFs are critical regulators of different adipocytes physiological processes such as autophagy, apoptosis, proliferation, and differentiation [16,17,18,19,20].

Hoxa5, also called Hox1.3, is a transcription factor and a member of the homeobox gene family [21,22]. Hoxa5 had been described to be involved in a number of different types of cancer, including small and non-small cell lung cancer, breast cancer, colorectal cancer, and so on. Hoxa5 was reported as downregulated in non-small cell lung cancer and could bind with the promoters of linc00312 to inhibit tumor proliferation [23]. It has also been shown that Hoxa5 counteracts stem cell traits through inhibiting Wnt signaling in colorectal cancer [24]. Meanwhile, previous studies also demonstrated that Hoxa5 is required for adipose tissue development [25] and is a positive regulator of adipogenesis both in humans and mice [26,27,28]. In 3T3-L1 preadipocyte, Hoxa5 influences adipogenesis by positive transcription regulation of the fatty-acid binding protein 4 (FABP4), which is the master adipogenesis regulator [29]. Moreover, Hoxa5 promoted the process of apoptosis through the Akt/mTORC1 signaling pathway in white adipose tissue (iWAT) of mice [30] and promoted polarization of M2 macrophages by activating the PPARγ pathway in mouse adipose tissue [31]. Overall, these various studies indicated the multi-functionality of Hoxa5 in different species. However, whether Hoxa5 can regulate differentiation and adipogenesis in goat subcutaneous preadipocytes is still unclear. Our previous study found that Hoxa5 gene expression is significantly changed during goat subcutaneous adipocyte differentiation. We hypothesized that Hoxa5 can regulate the differentiation and adipogenesis of goat subcutaneous preadipocyte. Thus, we explored potential mechanisms and biological roles of Hoxa5 on goat preadipocyte proliferation and differentiation in the current study.

Lipid deposition directly affects the meat quality of meat-producing animals that seriously affect meat-producing animal production efficiency. However, fat deposition in meat-producing animals involves a series of biological processes, such as preadipocyte differentiation and proliferation, and these biological processes are regulated by multiple transcription factors. The expansion of adipose tissue is increased when the excessive differentiation or proliferation of preadipocytes cause an increase of their size or numbers. Hoxa5, as a transcription factor, has already been shown as a positive regulator of adipogenesis both in mice and humans. However, the biological roles of Hoxa5 on goat preadipocyte proliferation and differentiation are still unknown. Hence, better elucidating the molecular mechanisms underlying goat preadipocyte proliferation and differentiation is helpful to increase the efficiency of livestock production.

## 2. Materials and Methods

### 2.1. Animals and Cell Culture

All experiments procedures involving animals were approved by the Institutional Animal Care and Use Committee of Southwest Minzu University (Chengdu, Sichuan, China). Goat subcutaneous preadipocytes isolation and culture methods were in accordance with previously described methods [32], and goat subcutaneous preadipocytes were isolated from the subcutaneous adipose tissues of Jianzhou Daer goats. Briefly, subcutaneous adipose tissues were isolated from three 7-day-old male Jianzhou Daer goats, and rinsed twice with 4 °C pre-cooling phosphate-buffered saline (PBS) containing antibiotics for cleaning. Then, goat subcutaneous adipose tissues were minced with scissors and digested with collagenase type 1 at 37 °C for 1 h. The digested samples were filtrated through mesh filters of 75-μm and centrifuged at 2000 r/min for 5 min, and then red blood cells (RBC) were disposed with RBC lysis buffer lysed solution. After that, the cell suspension was centrifuged at 1500 r/min for 5 min again and the pre-adipocytes were re-suspended in DMEM/F12 (Hyclone, Logan, UT, USA), supplemented with 10% FBS (Gemini, Calabasas, CA, USA) and 1% penicillin–streptomycin. Finally, the cell suspension was transferred to cell culture flask and incubated in an incubator with 5% CO_2_ at 37 °C.

### 2.2. Vectors Construction, Chemical Synthesis of siRNA, and Transfection

The full-length sequence of goat Hoxa5 (GenBank accession no. MZ004987) was amplified for the construction of the overexpressed Hoxa5 plasmid. After being processed with restriction enzymes (XhoI; Takara, Dalian, China; KpnI; Takara, Dalian, China) and T4 DNA Ligase (Takara, Dalian, China), the amplified sequences were ligated into pEGFP-N1 vector. Then, the recombinant plasmid was identified by enzyme digestion and DNA sequencing. All the plasmids were extracted by mini plasmid extraction kit (TIANGEN, Beijing, China) and stored at 4 °C.

For RNA interference experiments, Si-RNA targeting Hoxa5 (names as Si-Hoxa5) and a negative control Si-RNA (named as Si-NC) was designed and synthesized by Invitrogen. Sequences of the Si-RNA were as follows:

Si-Hoxa5: 5′-UGAAUUGCUCGCUCACGGAACUAUGdTdT-3′

Si-NC: 5′-UUCUCCGAACGUGUCACGUdTdT-3′.

All transfection experiments were performed with TurboFect Transfection Reagent (Thermo, Waltham, MA, USA) according to the manufacturer’s instructions. Taking 6-well-plate as an example, at 70–80% cell confluence in each well was transfected with 2 μg of plasmid DNA. For silencing Hoxa5, a volume of 6 μL Si-Hoxa5 or Si-NC in each well was used. The medium was exchanged after 24 h, and after 48 h of culture, the cells were collected for subsequent experimentation.

### 2.3. Induced Differentiation of Goat Subcutaneous Preadipocytes

For adipocyte differentiation, goat subcutaneous preadipocytes were expanded in culture using DMEM/F12 cell culture medium with 10% FBS and 1% antibiotic. The third-generation goat subcutaneous preadipocytes were seeded at 8 × 10^4^ cells/well in 6-well plates. After transfection for 24 h, goat subcutaneous preadipocytes that reached 70~80% confluence were cultured in an adipocyte-inducing medium (MEM/F12 containing 10% FBS, 1% antibiotic and 50 μmol·L^−1^ oleic acid (Sigma, Tokyo, Japan)) for 48 h.

### 2.4. Oil Red O Staining and Bodipy Staining

Adipogenic differentiation of goat subcutaneous preadipocytes was determined by Oil Red O staining or Bodipy staining, as we previously described [33]. Differentiated adipocytes were gently washed with PBS and fixed with 4% formaldehyde for 15 min followed by PBS wash. Then, the cells were stained with the Oil Red O or Bodipy working solutions for 20 min at room temperature. The cells were observed and photographed with an Olympus IX-73 epifluorescence microscope equipped (Tokyo, Japan) after staining and washing. For Oil Red O quantification, 1 mL 100% isopropanol was added to each well to extract the dye. The absorbance of extracted dye was then detected at 490 nm using an enzyme-labeled instrument.

### 2.5. MTT Analysis

MTT assay was used for the evaluation of goat subcutaneous preadipocyte proliferation. Goat subcutaneous preadipocytes were seeded in 96 well plates at a density of with 3 × 10^3^ cells per well. After 0, 24, 48, and 72 h of cell treatment, 10 μL MTT reagent (solarbio, Beijing, China) was added into each well away from light, then cells were incubated at 37 °C with 5% CO_2_ for 4 h. At last, the absorbance was measured using an enzyme-labeled instrument at a wavelength of 490 nm.

### 2.6. EdU Staining

The capacity of goat subcutaneous preadipocytes was measured by EdU incorporation assay (Beyotime, Shanghai, China). First, goat subcutaneous preadipocytes were seeded in 96-well plates, then they were transfected when the confluence reached 50%. At 48 h after transfection, goat subcutaneous preadipocytes were stained with 50 mM EdU solution for 2 h, fixed with 4% paraformaldehyde for 20 min, and permeated with 0.3% triton X-100 for 15 min. The cells were then incubated with Click Reaction Mixture provided by Edu staining Kit for 30 min in dark and stained with DAPI for 10 min. Finally, the cells were observed and imaged under an Olympus IX-73 epifluorescence microscope equipped.

### 2.7. RNA Isolation and Real-Time Quantitative PCR (qRT-PCR) Analysis

Total RNA was extracted with an RNAiso Plus (Takara, Dalian, China), according to the manufacturer’s instructions. cDNA was synthesized using the RevertAid First Strand cDNA Synthesis Kit (Thermo, Waltham, MA, USA), followed by amplification with the SYBR Green PCR Master Mix (Takara, Dalian, China). Specific primer sequences were designed as Table 1. qRT-PCR was carried out with a BioRad Real-Time PCR system. Data were analyzed using the comparative Ct method (2^−ΔΔCt^) [34].

### 2.8. Western Blot Analysis

Cells were washed twice with ice-cold PBS, scraped, and lysed in RIPA buffer containing protease inhibitor. Protein lysates (20 μg/lane) were subjected to 12% SDS-PAGE gel and then transferred to polyvinylidene fluoride membrane. Next, the membranes were blocked with 5% non-fat dry skim milk at room temperature for 2 h and were subsequently incubated with anti-Hoxa5 (1:500, Boaosen, Beijing, China, bs-5713R) and anti-β-actin (1:5000, Abways, Shanghai, China, AB0035) at 4 °C overnight. Membranes were washed in Tris-buffered saline with Tween 20 and incubated with horse radish peroxidase-conjugated goat anti-rabbit IgG secondary antibodies (1:5000, Abways, Shanghai, China, AB0102). Finally, target proteins were visualized by the enhanced chemiluminescence (ECL) detection systems (Thermo, Waltham, MA, USA).

### 2.9. Statistical Analysis

Statistical analyses were performed with GraphPad Prism 9 software. Two-tailed Student t-test was used to determine the significance of the indicated comparisons. Multiple comparison test was used to analyze the *p*-value in more than two groups. All values were presented as mean ± standard (SD) unless otherwise indicated. *p* values < 0.05 were considered statistically significant (* *p* < 0.05; ** *p* < 0.01).

## 3. Results

### 3.1. Hoxa5 Expression Changed during Goat Subcutaneous Preadipocytes Differentiation

Hoxa5, as a transcription factor, has been reported to regulate 3T3-L1 preadipocyte proliferation and differentiation [29,30]. Nevertheless, the role of Hoxa5 in goat subcutaneous preadipocytes has not been reported. In order to clarify the role of Hoxa5 in the differentiation of goat subcutaneous preadipocytes, we first isolated subcutaneous preadipocytes from goat subcutaneous adipose tissues and induced them to adipogenic differentiation. Oil Red O staining and qRT-PCR were used to ascertain the extent of goat subcutaneous preadipocytes differentiation. Our results showed that the number and size of the lipid droplets increased with the extension of induction time (Figure 1A). The expression level of adipocyte markers PPARγ, C/EBPα, and C/EBPβ showed a rising trend compared to 0 Day (Figure 1B–D). All the above indicated that goat subcutaneous preadipocytes were isolated successfully. Next, qRT-PCR was conducted to research the role of Hoxa5 in adipogenic differentiation of goat subcutaneous preadipocytes. Our result showed that Hoxa5 expression was significantly upregulated in differentiated cells, except for the cells in the third day of differentiation induction (Figure 1E). All the above data indicate that Hoxa5 may regulate the adipogenic differentiation of goat subcutaneous preadipocytes.

### 3.2. Hoxa5 Promotes Adipogenic Differentiation of Goat Subcutaneous Preadipocytes

To explore the role of Hoxa5 in goat subcutaneous preadipocytes, we first conducted gain-of-function studies in goat subcutaneous preadipocytes with pEGFP-Hoxa5 (named as Hoxa5) or pEGFP-N1 (named as Vector). Overexpression efficient of Hoxa5 in goat subcutaneous preadipocytes was validated by qRT-PCR and WB (Figure 2A,B, see Appendix A for all western original images). The expression of Hoxa5 in goat subcutaneous preadipocyte increased ~130 times, caused by Hoxa5 transfection rather than the same amount of control vector transfected cells (Figure 2A). PPARγ, C/EBPα, C/EBPβ, AP2, and SREBP1, which are considered to be classical differentiation markers in adipogenesis, were detected to assess whether adipocyte differentiation was altered by Hoxa5. The overexpression of Hoxa5 significantly increased PPARγ, C/EBPα, C/EBPβ, AP2, and SREBP1 mRNA level compared to control groups (Figure 2C). In addition, morphological observation, including Oil Red O staining and Bodipy staining, was performed. We found that not only was the extent of intracellular lipid accumulation augmented but the number of lipid droplets had increased, both of which were caused by overexpressed Hoxa5 expression (Figure 2D–F). These data suggested that overexpression of Hoxa5 can increase goat subcutaneous preadipocyte lipid accumulation.

We further validated the effect of Hoxa5 on adipogenic differentiation of goat subcutaneous preadipocytes. Si-NC or Si-Hoxa5 was transfected to goat subcutaneous preadipocytes to silence Hoxa5 expression. Hoxa5 interference efficiency was measured by qRT-PCR and WB, as shown in Figure 3A,B (See Appendix A for all western original images). Compared to Si-NC group with 48 h post-transfection, Si-Hoxa5 showed >40% knockdown efficiency. qRT-PCR results showed that Si-Hoxa5 significantly inhibited the expression of classical differentiation markers, such as PPARγ, C/EBPβ, AP2, and SREBP1 at 48 h post-differentiation, except C/EBPα (Figure 3C). Coincidently, Oil Red O signal and Bodipy staining showed that Si-Hoxa5 decreased intracellular lipid content and the number of lipid droplets in goat subcutaneous preadipocytes when compared to Si-NC (Figure 3D–F). Overall, these data indicated that Hoxa5 promotes adipogenic differentiation of goat subcutaneous preadipocyte.

### 3.3. Hoxa5 Suppresses the Proliferation of Goat Subcutaneous Preadipocytes

With the purpose of exploring the function of Hoxa5 in goat subcutaneous preadipocyte proliferation, crystal violet staining, MTT, qRT-PCR, and EdU staining were used to detect the proliferation of goat subcutaneous preadipocytes. At 24, 48, and 72 h after Hoxa5 transfection, goat subcutaneous preadipocytes were fixed using 4% formaldehyde and stained with crystal violet to indicate the relative number of cells. Violet staining assay analysis showed that the overexpression of Hoxa5 significantly inhibited the number of goat subcutaneous preadipocytes at any time point (Figure 4A). In compliance with MTT assay data, the Hoxa5 gain-of-function significantly decreased the OD value of 490 nm, which was proportional to cell number at 24, 48, and 72 h compared with the control group (Figure 4B). Further proliferation-related genes expression analysis showed Hoxa5 overexpression significantly reduced the mRNA levels of CCNE1, PCNA, CCND1, and CDK2, which were cell proliferation positively correlated markers (Figure 4C). Consistently, Hoxa5 overexpression significantly reduced the positive rate of EdU in goat subcutaneous preadipocytes compared with the cells transfected with Vector (*p* < 0.05, Figure 4D,E), indicating that Hoxa5 could inhibit the proliferative activity of goat subcutaneous preadipocytes.

Loss-of-function experiments were also performed to further determine the function of Hoxa5 in goat subcutaneous adipocyte proliferation. However, contrary to gain-of-function experiment, Si-Hoxa5 had the opposite effect. Crystal violet staining revealed an increase in the number of goat subcutaneous adipocytes when transfected with Si-Hoxa5 (Figure 5A). The activity of goat subcutaneous adipocytes treated with Si-Hoxa5 determined by MTT assay exhibited a similar increase trend to that of the crystal violet staining assay in Figure 5B. However, when compared with cells with gain-of-function of Hoxa5, results of the proliferation related gene expression assay revealed that inhibition of Hoxa5 only altered the expression of PCNA (Figure 5C). Moreover, EdU staining analysis also indicated that the number of EdU-labeled cells was significantly increased in Si-Hoxa5 group compared to that of the control (Figure 5D,E). Taken all together, the current evidence strongly suggests that Hoxa5 suppresses the proliferation of goat subcutaneous adipocytes proliferation.

### 3.4. Hoxa5 Affects Subcutaneous Preadipocytes Proliferation and Differentiation Possibly Targets the Promoter of PPARγ, SREBP1, and PCNA

Because Hoxa5 promotes goat subcutaneous preadipocyte differentiation and inhibits its proliferation by up- or downregulated several marker genes, we found that PPARγ, C/EBPβ, AP2, and SREBP1 expression was upregulated by overexpressed Hoxa5 during goat subcutaneous preadipocyte differentiation and PCNA was downregulated by overexpressed Hoxa5 during goat subcutaneous preadipocyte proliferation (Figure 2C and Figure 4C). In contrast, silence of Hoxa5 showed that PPARγ, C/EBPβ, AP2, and SREBP1 expression was downregulated and PCNA expression was upregulated (Figure 3C and Figure 5C). Thus, we speculated that PPARγ, C/EBPβ, AP2, SREBP1, and PCNA might be the potential target genes of Hoxa5. To validate this conjecture, we first analyzed Hoxa5 transcription binding motifs in genes promoter area (Figure 6A). Previous studies have shown that AP2, which is also known as FABP4, is a downstream target of Hoxa5. Thus, we investigated whether the other genes promoter (−2000~+100) contains Hoxa5 binding sequences. We found that there were 6 possible transcription binding sites in the promoter region of the PPARγ, 4 possible transcription binding sites in the promoter region of the SREBP1, and 6 possible transcription binding sites in the promoter region of the PCNA (Figure 6B–D). These data jointly suggest that Hxoa5 promotes goat subcutaneous preadipocyte differentiation and inhibits proliferation through targeting PPARγ, SREBP1, and PCNA expression.

## 4. Discussion

As transcription factors have been found to play an increasingly important regulatory role in the development and expansion of adipose tissue, many transcription factors such as C/EBPα, C/EBPβ, and PPARγ have been regarded as potential targets for obesity-related metabolic diseases [16,35]. Transcription factors specifically regulated the expression of numerous downstream target genes to involve in the regulation of adipogenesis, and this has become a popular research topic [14,15]. Hoxa5 is an important transcription factor. Nevertheless, little is known about the specific regulatory mechanisms and the role of Hoxa5 in regulating adipogenesis in livestock animals. In this study, we explored the effects of Hoxa5 on adipogenesis in goats, an important meat animal. Our previous study showed that Hoxa5 expression was significantly changed after induced differentiation of goat subcutaneous preadipocyte. Therefore, we conjectured that Hoxa5 may play a key regulatory role in adipogenesis in goats. Interestingly, we report that Hoxa5 is a positive regulator of adipocyte differentiation. Hoxa5 promotes the differentiation of adipocytes and inhibits the proliferation of goat subcutaneous preadipocytes (Figure 2, Figure 3, Figure 4 and Figure 5).

In this study, Hoxa5 promoted the differentiation of goat subcutaneous preadipocytes, which was characterized by increased lipid accumulation and upregulated adipocyte differentiation marker genes expression. At the molecular level, overexpression of Hoxa5 in goat subcutaneous preadipocytes resulted in upregulation of adipogenic genes, including PPARγ, C/EBPα, C/EBPβ, AP2, and SREBP1. At the morphological level, Hoxa5 overexpression could increase lipid droplet number and enhance the accumulation of lipids. This effect may be due, in part, to the higher expression of PPARγ, C/EBPα, C/EBPβ, AP2, and SREBP1. A recent study has also demonstrated that Hoxa5 was overexpressed in 3T3-L1 cell, and the mRNA levels of PPARγ, C/EBPβ, and FABP4 (AP2) were significantly enhanced, leading to lipid deposition [29]. Our results are now in agreement with the findings of the study. However, we found that CEBP/α expression upregulated in the same manner as the opposite two experimental conditions (overexpression and suppression of Hoxa5). This could be because the expression of genes in cells is regulated by multiple genes, and the intracellular signal regulation network is very complex. Ideally, overexpression and interference of a gene will present opposite downstream gene expression trend, but we found that some genes showed the same expression trend in the upstream molecular overexpression and interference cells. Therefore, we guessed that this may be attributed to the complexity of cellular signaling regulatory networks.

As is well known, the main cellular basis of adipogenic is the increased number of adipocytes and the increased lipid storage in adipocytes [16]. Therefore, we further determined the effects of Hoxa5 on goat subcutaneous preadipocyte proliferation. Hoxa5 gain-of-function inhibits the proliferation of goat subcutaneous preadipocyte, which is tested by three techniques (crystal violet staining, MTT assay, and EdU). The proliferation of cells is featured with genes expression alteration such as CCNE1, PCNA, CCND1, CDK2, and so on [36,37,38,39]. Indeed, Hoxa5 overexpression decreased preadipocyte proliferation, and CCNE1, PCNA, CCND1, and CDK2 are the representative downregulated genes (Figure 4C). It is interesting that the proliferation-related genes expression, except PCNA, in Hoxa5 knockdown goat subcutaneous preadipocytes are opposite with the result in Hoxa5 overexpression cells. Consistent with our findings, the overexpression of Hoxa5 in cervical cancer markedly inhibited cell proliferation through protein p27 and kinase B [40]. In cervical cancer, Hoxa5 inhibited the proliferation and tumor formation through repression the activity of Wnt/β-catenin pathway [41]. However, other studies have reported that knockdown of Hoxa5 suppressed the esophageal squamous cell carcinoma (ESCC) cell proliferation, which might be partly mediated via interfering with Wnt/β-catenin signaling pathway [42]. These experiments indicated that the specific molecular mechanisms and roles in cell proliferation may vary in different cell types.

The process of adipocyte differentiation is tightly and sequentially regulated by transcription factors which modulate the expression of numerous downstream target genes to generate model mature adipocyte phenotype [43]. For instance, the C/EBPβ and PPARγ were the major regulatory genes for adipogenesis that cooperatively control the expression of adipocyte-specific genes involved in the switch between differentiation and proliferation in preadipocytes [44,45]. Expression of FABP4 is known to be highly induced during adipocyte differentiation and transcriptionally controlled by PPARγ agonists [46]. Moreover, a previous study reported that Hoxa5 is a transcription factor which can affect the function of pathological scar-derived fibroblasts by transcriptionally upregulating the expression of p53 [47]. Interestingly, we found that among the markers of differentiation, only PPARγ and SREBP1 promoter regions have the binding sequence of Hoxa5 (Figure 6B,C). However, among the proliferation-related genes, only the trend of PCNA expression correspond in the cells that interfere with or overexpress Hoxa5. Therefore, we only analyzed the promoter region of PCNA, and the results showed that there were six potential binding sequences of Hoxa5 (Figure 6D). However, whether Hoxa5 could target PPARγ, SREBP1, and PCNA to regulate the differentiation and proliferation of goat subcutaneous preadipocyte remains to be further explored.

## 5. Conclusions

In conclusion, our data indicates that Hxoa5 promoted the adipogenic differentiation of goat subcutaneous preadipocytes and suppressed their proliferation may through targeting PPARγ, SREBP1, and PCNA expression. These results suggest that Hoxa5 is a new candidate gene in goat adipogenic differentiation.

## Figures and Tables

**Figure 1 animals-12-01859-f001:**
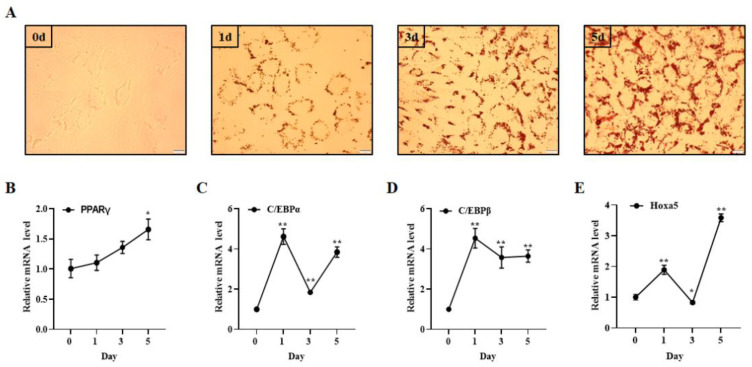
Hoxa5 expression changed during goat subcutaneous preadipocytes differentiation. (**A**) Representative images of Oil O staining of goat subcutaneous preadipocytes cultured in oleic acid (50 μmol·L^−1^) induction medium for different days. All images were examined under microscopy at a magnification of 400 × (scale bar = 20 um). (**B**–**D**) qRT-PCR analysis of the relative level of PPARγ, C/EBPα and C/EBPβ during preadipocyte adipogenic differentiation. (**E**) qRT-PCR analysis of the relative level of Hoxa5 expression in goat subcutaneous preadipocytes cultured in oleic acid (50 μmol·L^−1^) induction medium for the days. Each experiment was performed at least in triplicate, which yielded similar results. * indicates *p* values < 0.05 in comparison to 0 day group, ** indicates *p* values < 0.01 in comparison to 0 day group.

**Figure 2 animals-12-01859-f002:**
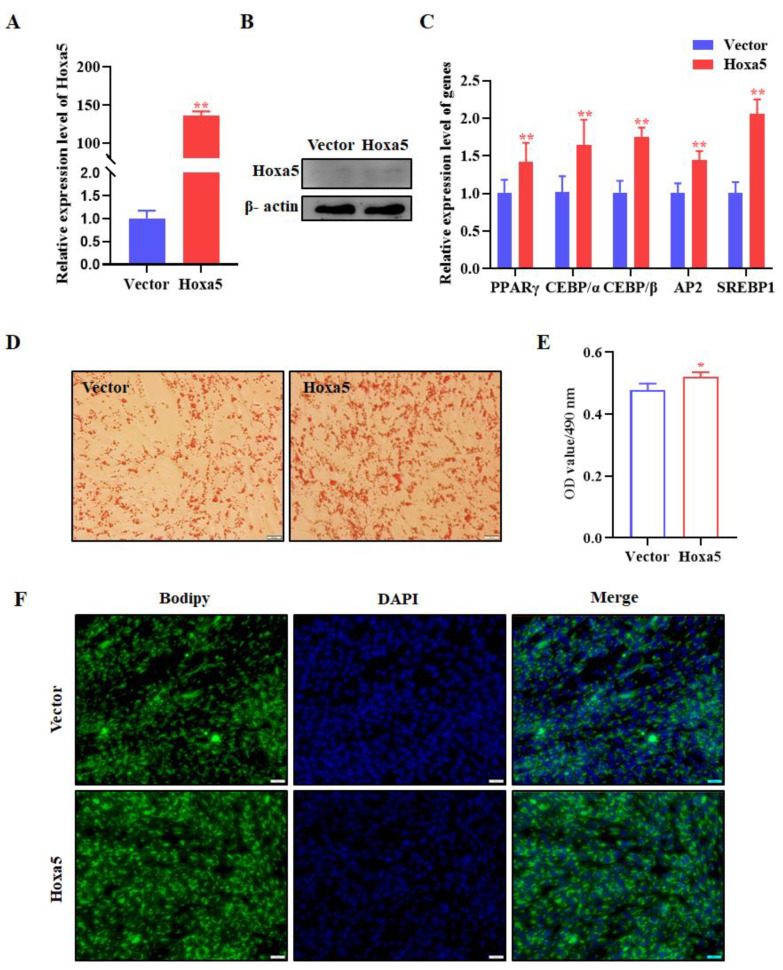
Overexpression of Hoxa5 promotes the adipogenic differentiation of goat subcutaneous preadipocytes. (**A**) qRT-PCR analysis of levels of Hoxa5 expression in goat subcutaneous preadipocytes with Hoxa5 or vector transfected for 48 h. (**B**) Hoxa5 protein levels and representative WB images. (**C**) qRT-PCR analysis of levels of gene expression in goat subcutaneous preadipocytes with Hoxa5 or vector. (**D**) Representative images (400×; scale bar = 20 um) of Oil Red O staining of goat subcutaneous preadipocytes with Hoxa5 or vector and (**E**) semi-quantitative assessment of Oil Red O content absorbance detection at 490 nm. (**F**) Representative images (400×; scale bar = 20 um) of mature adipocytes stained with Bodipy. Each experiment was performed at least in triplicate, which yielded similar results. * indicates *p* values < 0.05, ** indicates *p* values < 0.01.

**Figure 3 animals-12-01859-f003:**
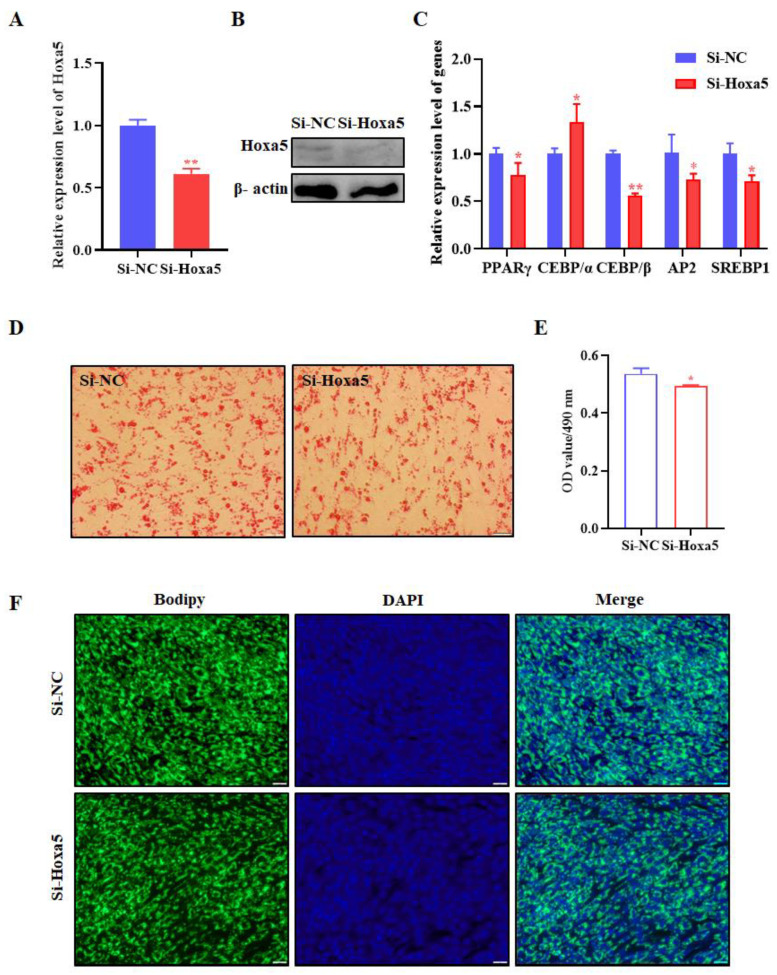
Silence of Hoxa5 inhibits the adipogenic differentiation of goat subcutaneous preadipocytes. (**A**) qRT-PCR analysis of levels of Hoxa5 expression in goat subcutaneous preadipocytes with Si-Hoxa5 or Si-NC transfected for 48 h. (**B**) Hoxa5 protein levels and representative WB images. (**C**) qRT-PCR analysis of levels of genes expression in goat subcutaneous preadipocytes with Si-Hoxa5 or Si-NC. (**D**) Representative images (400×; scale bar = 20 um) of Oil Red O staining of goat subcutaneous preadipocytes with Si-Hoxa5 or Si-NC and (**E**) semi-quantitative assessment of Oil Red O content absorbance detection at 490 nm. (**F**) Representative images (400×; scale bar = 20 um) of mature adipocytes stained with Bodipy. Each experiment was performed at least in triplicate, which yielded similar results. * indicates *p* values < 0.05, ** indicates *p* values < 0.01.

**Figure 4 animals-12-01859-f004:**
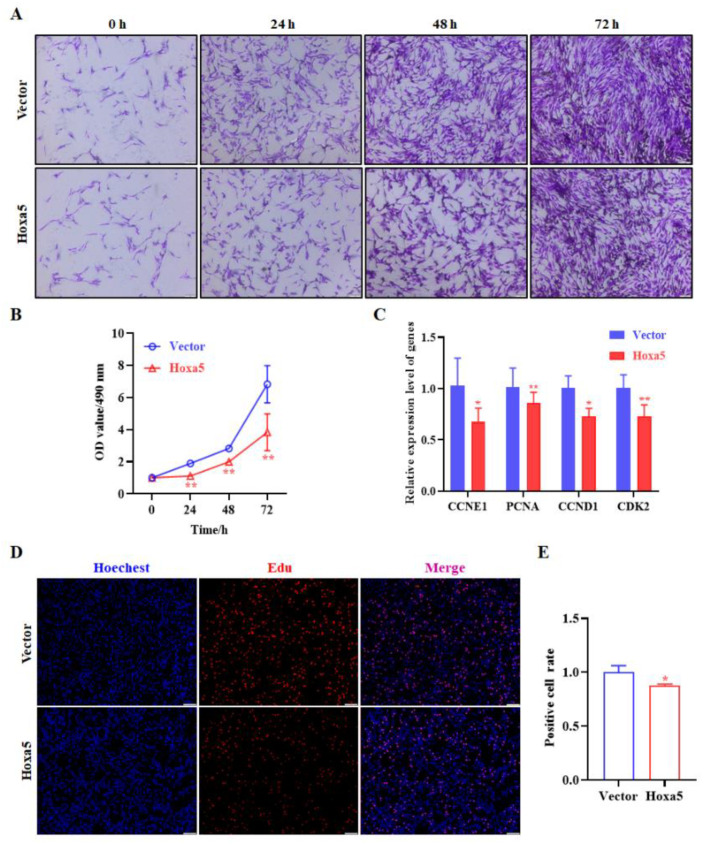
Overexpression of Hoxa5 inhibits goat subcutaneous preadipocytes proliferation. (**A**) The number of goat subcutaneous preadipocytes were determined using the crystal violet staining; 0, 24, 48, and 72 h after transfection cells were fixed, stained with crystal violet, and photographed (100×; scale bar = 100 um). (**B**) Cell proliferation was examined by MTT analysis. Cell proliferation was measured at 0, 24, 48, and 72 h. (**C**) The mRNA levels of CCNE1, PCNA, CCND1, and CDK2 were determined by qRT-PCR. (**D**) The proliferation capacity of goat subcutaneous preadipocyte was examined by the EdU assay, and representative images were examined under microscopy at a magnification of 100× (scale bar = 100 um). Proliferating cells were labeled with EdU (red). The nuclei were labeled by hoechst33342 (blue) and (**E**) The images were representative of the results obtained. Each experiment was performed at least in triplicate, which yielded similar results. * indicates *p* values < 0.05, ** indicates *p* values < 0.01.

**Figure 5 animals-12-01859-f005:**
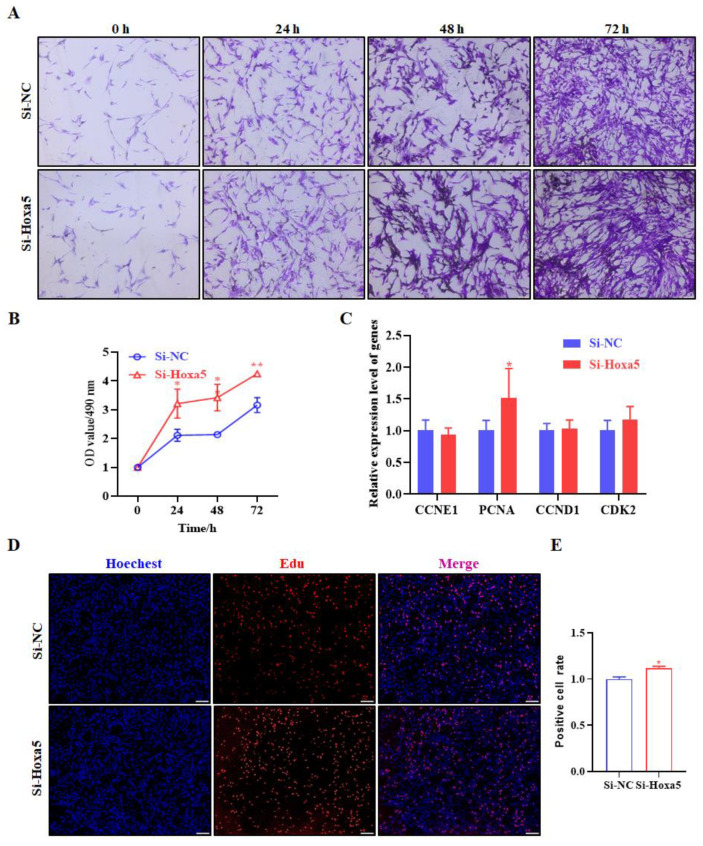
Silence of Hoxa5 promotes goat subcutaneous preadipocytes proliferation. (**A**) The number of goat subcutaneous preadipocytes were determined using the crystal violet staining; 0, 24, 48, and 72 h after transfection cells were fixed, stained with crystal violet, and photographed (100×; scale bar = 100 um). (**B**) Cell proliferation was examined by MTT analysis. Cell proliferation was measured at 0, 24, 48, and 72 h. (**C**) The mRNA levels of CCNE1, PCNA, CCND1, and CDK2 were determined by qRT-PCR. (**D**) The proliferation capacity of goat subcutaneous preadipocytes was examined by the EdU assay, and representative images were examined under microscopy at a magnification of 100× (scale bar = 100 um). Proliferating cells were labeled with EdU (red). The nuclei were labeled by Hoechst33342 (blue) and (**E**) The images were representative of the results obtained. Each experiment was performed at least in triplicate, which yielded similar results. * indicates *p* values < 0.05, ** indicates *p* values < 0.01.

**Figure 6 animals-12-01859-f006:**
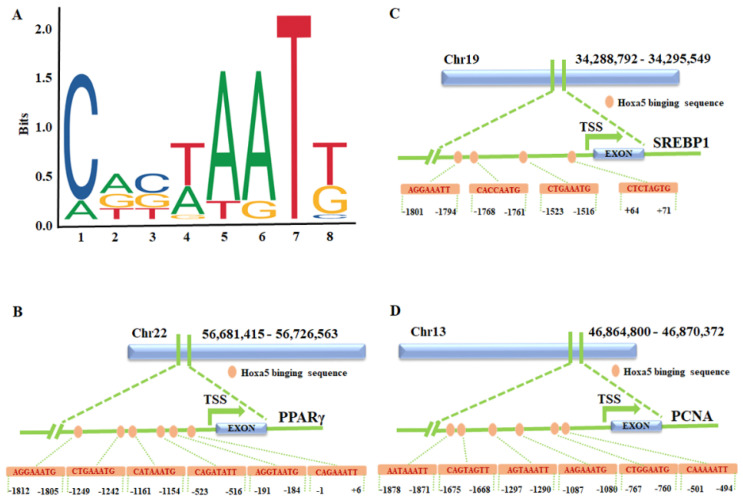
The effects of Hoxa5 on goat subcutaneous adipocyte proliferation and differentiation, possibly through the regulation of PPARγ, SREBP1, and PCNA. (**A**) The DNA-binding motif of Hoxa5 analyzed by Jasper. (**B**) The Hoxa5-binding sites prediction at the promoters of PPARγ, orange circles representing the Hoxa5-binding sites, blue boxes representing exons. (**C**) The Hoxa5-binding sites prediction at the promoters of SREBP1, orange circles represent the Hoxa5-binding sites, blue boxes represent exons. (**D**) The Hoxa5-binding sites prediction at the promoters of PCNA, orange circles represent the Hoxa5-binding sites, blue boxes represent exons.

**Table 1 animals-12-01859-t001:** Primer information for quantitative real-time PCR (qRT-PCR).

Gene Name	Forward Sequence (5′–3′)	Reverse Sequence (5′–3′)
Hoxa5	CTCATTTTGCGGTCGCTATC	ACGCTGAGATCCATGCCATT
C/EBPα	CTCCGGATCTCAAGACTGCC	CCCCTCATCTTAGACGCACC
C/EBPβ	CCGCCTTTAAATCCATGGAA	CTCGTGCTCTCCGATGCTAC
PPARγ	AAGCGTCAGGGTTCCACTATG	GAACCTGATGGCGTTATGAGAC
AP2	GTCCTTCAAATGGGCCAGGA	CTGGTGGTAGTGACACCGTT
SREBP1	AACATCTGTTGGAGCGAGCA	TCCAGCCATATCCGAACAGC
CCNE1	CTCCCTGATTCCCACACCTG	CATAAGATGCTTGTCCCTCA
PCNA	AGTGGAGAACTTGGAAAGGAA	GAGACAGTGGAGTGGCTTTTGT
CCND1	TGAACTACCTGGACCGCT	CAGGTTCCACTTGAGTTTGT
CDK2	GCCAGGAGTTACTTCTATGC	TGGAAGAAAGGGTGAGCC
UXT	GCAAGTGGATTTGGGCTGTAAC	ATGGAGTCCTTGGTGAGGTTGT

## Data Availability

Data are contained within the article.

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
