# Peer review of "Hoxa5 Inhibits the Proliferation and Induces Adipogenic Differentiation of Subcutaneous Preadipocytes in Goats"

_animals, 2022, doi:10.3390/ani12141859_

Round 1
Reviewer 1 Report
In this manuscript, Chen et al. investigate the role of Hoxa5 in the regulation of goat subcutaneous preadipocyte differentiation and proliferation. They use Hoxa5 gain of function and loss of function approaches in a goat subcutaneous preadipocyte tissue culture system.
Three main findings are presented:
i. Overexpression of Hoxa5 increased expression of adipogenic genes and promoted subcutaneous preadipocyte lipid accumulation. Hoxa5 overexpression also reduced cell proliferation, as measured by crystal violet staining, MTT assay, marker gene expression, and EdU staining.
ii. Knockdown of Hoxa5 by siRNA inhibited differentiation of subcutaneous preadipocytes and suppressed the formation of lipid droplets. Hoxa5 knockdown increased cell proliferation, as measured by crystal violet staining, MTT assay, and EdU staining.
iii. Expression of Hoxa5 was altered in a mouse DMD model: Hoxa5 mRNA levels are lower in the subcutaneous and visceral fat of the mutant than in WT, similar to other adipogenic transcription factors.
The claims are well supported by the data, and I recommend it being published, subject to the authors addressing a few minor concerns.
Minor points:
i. Lines 71-72. This sentence has strange phrasing, which reads as though the authors are claiming to have discovered DMD. Please change this.
ii. Some of the figures include scale bars (e.g. Fig2E) but these are not explained in figure legends. Please state the size of the scale bars in the figure legends.
iii. Fig1E. The graph showing the 0 d vs 3 d differentiation time has a y-axis that starts at 0.7, while the other graphs start at 0.0. This makes the graph look misleading. Please alter this so that each graph has a y-axis that starts at 0.0 for better comparison.
iv. Section 3.4 and Fig6. Hoxa5 could bind to distal enhancer elements of these genes, not just to the promoter regions. Please acknowledge this in the text.
Author Response
- Lines 71-72. This sentence has strange phrasing, which reads as though the authors are claiming to have discovered DMD. Please change this.
Response: Thank you very much for your valuable comment and suggestion. It has been reported that duchenne muscular dystrophy (DMD) is a muscle-wasting disease for which no cure exists caused by mutations in the DMD gene (Lala-Tabbert, N.; et al. 2016), and the text has been revised accordingly (line 76 to line 78).
- Some of the figures include scale bars (e.g. Fig 2E) but these are not explained in figure legends. Please state the size of the scale bars in the figure legends.
Response: Thank you very much for your valuable comment and suggestion. The images of Oil red O and Bodipy staining used in our study were obtained at a magnification of 400× (scale bar = 20 um), and the images of crystal violet staining were obtained at a magnification of 100× (scale bar = 100 um). Scale bars had been supplemented in corresponding figure legends (line 217 to line218; line 260 to line 263; line 270 to line 273; line 316 to line 317 and line 327 to line 328).
- Figure 1E. The graph showing the 0 d vs 3 d differentiation time has a y-axis that starts at 0.7, while the other graphs start at 0.0. This makes the graph look misleading. Please alter this so that each graph has a y-axis that starts at 0.0 for better comparison.
Response: Thank you very much for your valuable comment and suggestion. In order to better present our results, we changed the figure style in Fig 1E to Fig 1B, C and D.
- Section 3.4 and Fig6. Hoxa5 could bind to distal enhancer elements of these genes, not just to the promoter regions. Please acknowledge this in the text.
Response: Thank you very much for your valuable comment and suggestion. In our study, we found that PPARγ, C/EBPβ, AP2 and SREBP1 expression was upregulated by overexpressed Hoxa5 during goat subcutaneous preadipocyte differentiation and PCNA was downregulated by overexpressed Hoxa5 during goat subcutaneous preadipocyte proliferation (Figure 2C, 4C). In contrast, silence of Hoxa5 showed the opposite tendency (Figure 3C and 5C). Thus, we speculated that PPARγ, C/EBPβ, AP2, SREBP1 and PCNA might be the potential target genes of Hoxa5. To validate this conjecture, we first analyzed Hoxa5 transcription binding motifs in genes promoter area (Figure 6A). And the text has been revised accordingly (line 316, line 318 to line 328).
Reviewer 2 Report
Authors demonstrated that “Hoxa5 promotes adipogenic differentiation of goat subcutaneous preadipocyte and inhibits its proliferation in vitro, and Hoxa5 may positively regulate adipose tissue differentiation in vivo”. Several studies have already shown the role of HOXA5 as a positive regulator of adipogenesis both in human and mice, and its role in adipose tissue dysfunction, as demonstrated by PMIDs: 35623098, 35203377 and 26980478 (references that should be added to the manuscript of DingShuang Chen et al.), therefore the only novelty of this paper concerns the involvement of Hoxa5 as key player of in vitro adipogenesis in goat. This, however, leads to the important question of what is the rationale of this work. In particular, how Hoxa5 gene could impact on (or affect) the distribution of adipose tissue of meat-producing animals, including goat? Or may Hoxa5 represent a marker (or an indicator) of the meat quality of meat-producing animals? Why is this work important for the scientific community? Maybe for the meat-producing animal research? Trying to explain this would allow the reader to appreciate your work and to make the article more appealing to the readers of this journal.
Then, I struggle to understand the choice of validating the goat in vitro data with a mouse model of Duchenne muscular dystrophy (DMD), where whole adipose tissue was used instead of pre-adipocyte samples as seen in goat experiments. I believe DMD murine model is not a suitable model for the purpose of this study.
Therefore, if it is still possible, I would suggest to treat cultures of pre-adipocytes from goat with free fatty acids, which have been demonstrated to induce insulin resistance in adipocyte, mimicking what happens in obesity setting. This approach would appear more appropriate to understand the direct mechanism(s) between Hoxa5 gene and the dysfunction of adipose tissue of goat. Otherwise, it would be enough to discuss in more details the above mentioned published papers. This would render the study more impactful and give more robustness to the findings.
Minor points:
a) Regarding the paragraph 2.1, I would suggest to describe the goat subcutaneous pre-adipocyte isolation first and then the tissue collection from the DMD murine model;
b) Please add more details about induction of differentiation of goat subcutaneous pre-adipocytes;
c) Then, please check the qPCR primer sequences for Hoxa5, PPARγ, AP2, SREBP1, CCNE1, CDK2 and UXT genes, they do not seem perfectly aligned on the reference genome;
d) Student t test is not suitable when you have more than two groups. In fact, in this case it would be advisable to use multiple comparison tests;
e) data in figure 1E should be showed as in Figure 1B, C and D;
f) please show the WB analysis for quantify hoxa5 protein levels upon its overexpression and silencing in goat pre-adipocytes;
g) In the figure 3E and 4D, there are no clear differences between control and Hoxa5 silenced/overexpressed cells, respectively.
h) lack of literature about the link between Hoxa5 function and Wnt signalling pathway (PMID: 35203377 and PMID: 26678341) as well as the potential regulation of PPARγ by Hoxa5 (PMID: 31441588).
Author Response
1. Several studies have already shown the role of HOXA5 as a positive regulator of adipogenesis both in human and mice, and its role in adipose tissue dysfunction, as demonstrated by PMIDs: 35623098, 35203377 and 26980478 (references that should be added to the manuscript of DingShuang Chen et al.). This, however, leads to the important question of what is the rationale of this work. In particular, how Hoxa5 gene could impact on (or affect) the distribution of adipose tissue of meat-producing animals, including goat? Or may Hoxa5 represent a marker (or an indicator) of the meat quality of meat-producing animals? Why is this work important for the scientific community? Maybe for the meat-producing animal research? Trying to explain this would allow the reader to appreciate your work and to make the article more appealing to the readers of this journal.
Response: Thank you very much for your valuable comments and suggestions. References that have been added to the manuscript (line 61 to line 63); Lipid deposition directly affects the meat quality of meat-producing animals that seriously affect meat-producing animal production efficiency. However, fat deposition in meat-producing animals involve a series of biological processes, such as preadipocyte differentiation and proliferation, and these biological processes are regulated by multiple transcription factors. The expansion of adipose tissue is increased when the excessive differentiation or proliferation of predipocytes cause an increase of their size or numbers. Hoxa5 as a transcription factor have already been shown as a positive regulator of adipogenesis both in mice and human. However, the biological roles of Hoxa5 on goat preadipocyte proliferation and differentiation are still unknown. Hence, better elucidating of molecular mechanism underlying goat preadipocyte proliferation and differentiation is helpful to increase the efficiency of livestock production (line 82 to line 92).
2. Then, I struggle to understand the choice of validating the goat in vitro data with a mouse model of Duchenne muscular dystrophy (DMD), where whole adipose tissue was used instead of pre-adipocyte samples as seen in goat experiments. I believe DMD murine model is not a suitable model for the purpose of this study.
Response: Thank you very much for your valuable comments. Our in vitro study indicated that Hoxa5 could promote the differentiation of goat subcutaneous preadipocytes, and its expression up-regulated in the mature adipocytes, then we want to validate this in vivo. Because papers reported that intramuscular adipose tissue formation in a pathological condition like DMD, so we take the adipose tissue from DMD mice to study the correlation of Hoxa5 expression and the differentiation marker genes. Interestingly, the results in DMD mice adipose tissue were consistent with that in goat preadipocytes. Then, we thought Hoxa5 may be a maker of adipocyte differentiation.
3. Therefore, if it is still possible, I would suggest to treat cultures of pre-adipocytes from goat with free fatty acids, which have been demonstrated to induce insulin resistance in adipocyte, mimicking what happens in obesity setting. This approach would appear more appropriate to understand the direct mechanism(s) between Hoxa5 gene and the dysfunction of adipose tissue of goat. Otherwise, it would be enough to discuss in more details the above mentioned published papers. This would render the study more impactful and give more robustness to the findings.
Response: Thank you very much for your valuable suggestions. Our group have studied the deposition of lipid for a long time, and this is also the research area of our group. In the following study, we will try the model you provided to verify our in vitro data, we believe that the construction of this model will greatly improve the significance of our research and the quality of our articles.
Minor points:
a) Regarding the paragraph 2.1, I would suggest to describe the goat subcutaneous pre-adipocyte isolation first and then the tissue collection from the DMD murine model;
Response: Thank you very much for your valuable suggestion. We have describe the goat subcutaneous pre-adipocyte isolation first and then the tissue collection from the DMD murine model in paragraph 2.1, and the text has been revised accordingly (line 112 to line 115).
b) Please add more details about induction of differentiation of goat subcutaneous pre-adipocytes;
Response: Thank you very much for your valuable suggestion. For adipocyte differentiation, goat subcutaneous preadipocytes were expanded in culture using DMEM/F12 cell culture medium with 10% FBS and 1% antibiotic. The third-generation goat subcutaneous preadipocytes were seeded at 8 × 104 cells/well in 6-well plates. After transfection for 24 h, goat subcutaneous preadipocytes that reached 70%~80% confluence were cultured in an adipocyte inducing medium (MEM/F12 containing 10% FBS, 1% antibiotic and 50 μmol•L-1 oleic acid (Sigma) for 48 h. This section has been added to the method and the text has been revised accordingly (line 136 to line 141).
c) Then, please check the qPCR primer sequences for Hoxa5, PPARγ, AP2, SREBP1, CCNE1, CDK2 and UXT genes, they do not seem perfectly aligned on the reference genome;
Response: Thank you very much for your valuable comment. The qPCR primer sequences have been checked carefully, we have changed the wrong primer sequences, and have been revised accordingly in table 1.
d) Student t test is not suitable when you have more than two groups. In fact, in this case it would be advisable to use multiple comparison tests;
Response: Thanks very much for your suggestion, multiple comparison tests was used to analyze the p-value in more than two groups. In figure 1B-1E, * indicates p values < 0.05 in comparison to 0 day group, ** indicates p values < 0.01 in comparison to 0 day group. This section has been added to the legend of Figure 1 (line 222 to 223).
e) data in figure 1E should be showed as in Figure 1B, C and D;
Response: Thank you very much for your valuable suggestion. Figure 1E has been shown as in Figure 1B, C and D.
f) please show the WB analysis for quantify hoxa5 protein levels upon its overexpression and silencing in goat pre-adipocytes;
Response: Thank you very much for your valuable suggestion. WB analysis for quantify Hoxa5 protein levels upon its overexpression and silencing in goat pre-adipocytes have been down, and the results were consistent with the qPCR validation results as shown in figure 2B and 3B.
g) In the figure 3E and 4D, there are no clear differences between control and Hoxa5 silenced/overexpressed cells, respectively.
Response: Thank you very much for your valuable comment. Both the results were repeated for 3 times at least, and each time showed the same result that Hoxa5 promotes differentiation and inhibits proliferation of goat subcutaneous preadipocytes. For the figure 3E and 4D, there were obvious differences between control and Hoxa5 silenced/overexpressed cells before merge, and the quantity result also indicated the difference, maybe the Hoechst staining too strong to see the differences between the two groups.
h) lack of literature about the link between Hoxa5 function and Wnt signalling pathway (PMID: 35203377 and PMID: 26678341) as well as the potential regulation of PPARγ by Hoxa5 (PMID: 31441588).
Response: Thank you very much for your valuable suggestion. The references have been added and have been revised accordingly. (line 60 to line 63 and line 67 to line 68).
Reviewer 3 Report
This study aimed to investigate roles of Hoxa5 in goat adipogenic cells obtained from subcutaneous fat. The in vitro experiments were well designed and conducted to demonstrate functions of Hoxa5 in the goat cell culture.
However, there are some major concerns described below. The authors should address these issues before reviewer’s consideration for acceptance of this manuscript.
Major points:
The results of experiments using DMD mouse model is considerably detached from the original objective of this study. I do not understand what the authors mean by incorporating these data in this manuscript. DMD mouse has so different genetic background from goat, namely DMD phenotype and difference in animal species. Accordingly, the data from DMD mouse experiments should not be combined with those of goat cell culture experiments. The authors should remove the results and discussion regarding DMD mouse from this manuscript.
Minor points:
1. Why was CEBPA expression upregulated in the same manner in the opposite two experimental conditions (overexpression and suppression of Hoxa5) in goat cells? Please discuss.
2. Line 187, 189: correct the unit of oleic acid.
3. Line 193, 249, and others: gain-of-function
4. Line 204: We
5. Line 295-297: this sentence is awkward. Needs to be revised.
6. Line 300-301: this sentence has nonsense. Needs to be revised. What is opposite tendency? Please explain.
Author Response
1. The results of experiments using DMD mouse model is considerably detached from the original objective of this study. I do not understand what the authors mean by incorporating these data in this manuscript. DMD mouse has so different genetic background from goat, namely DMD phenotype and difference in animal species. Accordingly, the data from DMD mouse experiments should not be combined with those of goat cell culture experiments. The authors should remove the results and discussion regarding DMD mouse from this manuscript.
Response: Thank you very much for your valuable comments and suggestions. Our in vitro study indicated that Hoxa5 could promote the differentiation of goat subcutaneous preadipocytes, and its expression up-regulated in the mature adipocytes, then we want to validate this in vivo. Because papers reported that intramuscular adipose tissue formation in a pathological condition like DMD, so we take the adipose tissue from DMD mice to study the correlation of Hoxa5 expression and the differentiation marker genes. Interestingly, the results in DMD mice adipose tissue were consistent with that in goat preadipocytes. Then, we thought Hoxa5 may be a maker of adipocyte differentiation. So we did not remove this result, and will do further studies to explore the role of Hoxa5 in adipocyte differentiation.
Minor points:
1. Why was CEBPA expression upregulated in the same manner in the opposite two experimental conditions (overexpression and suppression of Hoxa5) in goat cells? Please discuss.
Response: Thank you very much for your valuable comment and suggestion. The expression of genes in cells is regulated by multiple genes, and the intracellular signal regulation network is very complex. Ideally, overexpression and interference of a gene will present opposite downstream gene expression trend, but we found that some genes showed the same expression trend in the upstream molecular overexpression and interference cells. This may be attributed to the complexity of cellular signaling regulatory networks. And the text has been revised accordingly (line 428 to 436).
2. Line 187, 189: correct the unit of oleic acid.
Response: Thanks very much for your valuable suggestion. The unit of oleic acid has been corrected (line 217, line 220 to line 221).
3. Line 193, 249, and others: gain-of-function.
Response: Thank you very much for your valuable suggestion. We have checked all the text and corrected the errors (line 226, 284).
4. Line 204: We
Response: Thank you for your valuable suggestion. We have checked all the text and corrected the same errors (line 236).
5. Line 295-297: this sentence is awkward. Needs to be revised.
Response: Thank you for your valuable suggestion. Based on the fact that Hoxa5 promotes goat subcutaneous preadipocyte differentiation and inhibits its proliferation by up or down regulated several marker genes. And the text has been revised accordingly (line 332 to line 333)
6. Line 300-301: this sentence has nonsense. Needs to be revised. What is opposite tendency? Please explain.
Response: Thank you for your valuable comment and question. We want to express the opposite regulation role of Hoxa5 overexpression and interference to the differentiation or proliferation markers. Maybe the representation is not clear, so we changed the sentence: In contrast, silence of Hoxa5 showed that PPARγ, C/EBPβ, AP2 and SREBP1 expression was downregulated and PCNA expression was upregulated. And the text has been revised accordingly (line 337 to line 339).
Round 2
Reviewer 2 Report
just one more technical point:
Please also add the corrections of statistical method used also in the paragraph 2.9 Statistical Analysis, not only to the figure legends:
Response: Thanks very much for your suggestion, multiple comparison tests was used to analyze the p-value in more than two groups. In figure 1B-1E, * indicates p values < 0.05 in comparison to 0 day group, ** indicates p values < 0.01 in comparison to 0 day group. This section has been added to the legend of Figure 1 (line 222 to 223).
Author Response
1、just one more technical point:
Please also add the corrections of statistical method used also in the paragraph 2.9 Statistical Analysis, not only to the figure legends.
Response: Thank you very much for your valuable comment and suggestion. Multiple comparison test was used to analyze the p-value in more than two groups. This section has been added to the paragraph 2.9 Statistical Analysis (line 179).
Reviewer 3 Report
I have to say again that the DMD mouse experiments should be deleted in this manuscript, otherwise I cannot recommend publication of this study any more.
The results of experiments using DMD mouse model is considerably detached from the original objective of this study. I do not understand what the authors mean by incorporating these data in this manuscript. DMD mouse has so different genetic background from goat, namely DMD phenotype and difference in animal species. Accordingly, the data from DMD mouse experiments should not be combined with those of goat cell culture experiments. The authors should remove the results and discussion regarding DMD mouse from this manuscript.
Author Response
1. I have to say again that the DMD mouse experiments should be deleted in this manuscript, otherwise I cannot recommend publication of this study any more.
The results of experiments using DMD mouse model is considerably detached from the original objective of this study. I do not understand what the authors mean by incorporating these data in this manuscript. DMD mouse has so different genetic background from goat, namely DMD phenotype and difference in animal species. Accordingly, the data from DMD mouse experiments should not be combined with those of goat cell culture experiments. The authors should remove the results and discussion regarding DMD mouse from this manuscript.
Response: Thank you very much for your valuable comment and suggestion. The results and discussions regarding DMD mouse have been removed from this manuscript. And the text has been revised accordingly.